# Find What You Want: Learning Demand-conditioned Object Attribute Space for Demand-driven Navigation

**Hongcheng Wang**
CFCS, School of CS, Peking University
`whc.1999@pku.edu.cn`

**Andy Guan Hong Chen**
School of EECS, Peking University
`cghandy@pku.edu.cn`

**Xiaoqi Li**
CFCS, School of CS, Peking University
`clorisli@stu.pku.edu.cn`

**Mingdong Wu**
CFCS, School of CS, Peking University
`wmingd@pku.edu.cn`

**Hao Dong**[*]
CFCS, School of CS, Peking University
`hao.dong@pku.edu.cn`

## Abstract

The task of Visual Object Navigation (VON) involves an agent's ability to locate a particular object within a given scene. To successfully accomplish the VON task, two essential conditions must be fulfiled: 1) the user knows the name of the desired object; and 2) the user-specified object actually is present within the scene. To meet these conditions, a simulator can incorporate predefined object names and positions into the metadata of the scene. However, in real-world scenarios, it is often challenging to ensure that these conditions are always met. Humans in an unfamiliar environment may not know which objects are present in the scene, or they may mistakenly specify an object that is not actually present. Nevertheless, despite these challenges, humans may still have a demand for an object, which could potentially be fulfilled by other objects present within the scene in an equivalent manner. Hence, this paper proposes Demand-driven Navigation (DDN), which leverages the user's demand as the task instruction and prompts the agent to find an object which matches the specified demand. DDN aims to relax the stringent conditions of VON by focusing on fulfilling the user's demand rather than relying solely on specified object names. This paper proposes a method of acquiring textual attribute features of objects by extracting common sense knowledge from a large language model (LLM). These textual attribute features are subsequently aligned with visual attribute features using Contrastive Language-Image Pre-training (CLIP). Incorporating the visual attribute features as prior knowledge, enhances the navigation process. Experiments on AI2Thor with the ProcThor dataset demonstrate that the visual attribute features improve the agent's navigation performance and outperform the baseline methods commonly used in the VON and VLN task and methods with LLMs. The codes and demonstrations can be viewed at https://sites.google.com/view/demand-driven-navigation.

---

[*]Corresponding author

37th Conference on Neural Information Processing Systems (NeurIPS 2023).

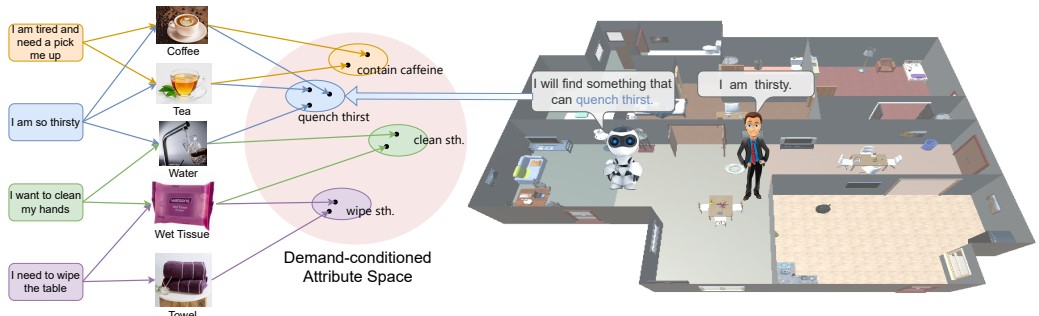

Figure 1: **Demand-driven Navigation and Demand-conditioned Attribute Space.** The left side shows a many-to-many mapping between demands and objects, *i.e.*, a demand can be satisfied by more than one object, and an object can satisfy more than one demand; but in any case, objects that can satisfy the same demand will have similar attributes. On the right side, a user at home provides the agent with an instruction "I am so thirsty". An agent determines what objects can satisfy this user's demand. Although the user does not indicate which specific object they want, the agent interprets that the object he wants must have the attribute of being able to quench thirst.

# 1    Introduction

Visual Object Navigation (VON) [1–15], which is considered fundamental for home service agents, refers to the task in which an agent is required to find a user-specified object within a provided scene. Previous VON studies can be broadly categorised into two groups. The first category, known as closed-vocabulary navigation [1–4, 6–10], involves a predetermined set of target object categories so that the agent is restricted to selecting objects from this predefined set. On the other hand, open-vocabulary navigation [5, 11, 12] encompasses a more flexible approach where the target objects are described in natural language, meaning the agent is not limited to specific predefined categories.

In real-world applications of VON, users are typically required to provide the name of the object they want the agent to locate, regardless of whether it falls under the closed-vocabulary or open-vocabulary group. This requirement introduces two conditions that must be fulfilled: 1) the user must possess knowledge of the object's name in the given environment; and 2) the specified object exists within the environment. These conditions are necessary for successful object navigation tasks in practical scenarios. In common simulators (such as AI2Thor [16] and Habitat [17, 18]), these two conditions can be easily satisfied. As direct access to the scene's metadata is available in the simulators, the existence of target objects can be checked using the metadata, or the object names in the metadata can be directly used as target objects. However, in the real world, where the user may not be omniscient of the environment (*e.g.*, at a friend's house), the two conditions described above may not be met, which will result in failure to satisfy the user's demand for an object. Additionally, it is possible that when the user specifies an object to search for the agent is able to find an object with a similar function within the scene, when the specified object does not exist. For example, an object such as a bottle of water may be specified which has a related demand, to quench thirst. In the case where a bottle of water is not present in the scene, this demand can be adequately met by a cup of tea or apple juice found by the agent, however this would fail the VON task. This challenge illustrates a key difficulty VON faces in real environments.

From a psychological point of view [19–22], this paper considers the user's motivation to make the agent search for an object. In the VON task the user initially has a certain demand, then thinks of an object to satisfy it, and finally asks the agent to search for that specific object. Then, would it not be more consistent with the constraints of the real world environment to omit the second step and directly use the demand as an instruction to ask the agent to find an object that satisfies the demand? This paper argues that utilising user demands as instructions has the following benefits: 1) Users no longer need to consider the objects in the environment, but only their own demands. This demand-driven description of the task is more in line with the psychological perspective of human motivations. 2) The mapping between demands and objects is many-to-many (*i.e.*, a demand instruction can be satisfied by multiple objects, and an object can potentially satisfy multiple demand instructions), which can theoretically increase the probability of an agent satisfying user demands in

comparison to VON. 3) The demands, when described in natural language have a wider descriptive space than just specifying an object, and the tasks are thus described more naturally and in accordance with the daily habits of humans.

Therefore, this paper proposes the task of Demand-Driven Navigation (DDN). In the DDN task, the agent receives a demand instruction described in natural language (*e.g.*, I am thirsty), and then the agent must search for an object within the provided environment to satisfy this demand. Our task dataset is generated semi-automatically: it is initially generated by GPT-3 [23] (In the entire paper, we use gpt-3.5-turbo version), and then manually filtered and supplemented. The DDN task presents new challenges for embodied agents: 1) Different real-world environments contain different objects and, even within a fixed environment, the categories of objects can change over time due to human interaction with the environment, resulting in agents potentially looking for different objects when attempting to satisfy the same demand. 2) The many-to-many mapping between instructions and objects requires agents to reason depending on common sense knowledge, human preferences and scene grounding information: to identify what objects potentially exist in the scene to satisfy the user's demand, and where these objects are most likely to be located. 3) The agent needs to determine whether the current objects within the field of view satisfy the user's demand from the objects' visual geometry. There exists the possibility that this reasoning is not only based on the visual features, but stems from the functionality of the object, necessitating the agent having common sense knowledge about the object. In summary, the core challenge of DDN is how to use **common sense knowledge**, **human preferences** and **scene grounding information** to **interpret** demand instructions and efficiently **locate** and **identify** objects that satisfy the user's demands.

It is noted that the essence of an object satisfying a demand is that certain attributes of the object fulfill the demand instructions. For example, for the demand of "I am thirsty," the object to be sought only needs to have attributes such as "potable" and "can quench thirst". Both a bottle of water and a cup of tea are objects that have these attributes. To address the challenge of mapping attributes and demand instructions while considering their relevance to the current scene, this paper proposes a novel method that extracts the common sense knowledge from LLMs to learn the demand-conditioned textual attribute features of objects. Then the proposed method aligns these textual attribute features with visual attribute features using the multi-modal model CLIP [24]. The learned visual attribute features contain common sense knowledge and human preferences from LLMs, and also obtain scene grounding information via CLIP. This paper also trains a demand-based visual grounding model to output the bounding box of objects in the RGB input that match the demand when an episode ends. Experiments are conducted on the AI2Thor simulator [16] with the ProcThor dataset [25]. The proposed method is compared with closed-vocabulary object navigation, open-vocabulary object navigation, and their variants. This paper also evaluates the performance of GPT-3 and multi-modal large language model(MM-LLM), MiniGPT-4 [26], operated agents which accomplish this task via prompt engineering. The experiments demonstrate that the demand-conditioned attribute features extracted from the LLM and aligned with CLIP effectively assist in navigation and outperform baselines. In summary, this paper's contributions are listed as follows:

- This paper proposes the task of Demand-Driven Navigation (DDN), that requires the agent to find an object to meet the user's demand. The DDN task relies heavily on common sense knowledge, human preferences, and scene grounding information.

- This paper evaluates two different categories of VON algorithms and their variants combined with GPT-3, as well as the performance of large language models (*e.g.*, GPT-3 and MiniGPT-4) on the DDN task. The results demonstrate that existing algorithms have difficulty solving the DDN task.

- This paper provides a novel method to tackle the DDN task and a benchmark for this. The proposed method extracts the common sense knowledge from LLMs to learn textual attribute features and uses CLIP to align the textual and visual attribute features. The experiments demonstrate that the demand-conditioned attribute features extracted from the LLM and aligned with CLIP effectively assist in navigation and outperform the baselines.

## 2 Related Work

### 2.1 Visual Navigation

The task of visual navigation requires the agent to use visual information [27, 28] to reach a target location, such as visual object navigation (VON) [1–15, 29], visual language navigation (VLN) [30–38], and visual audio navigation (VAN) [39–44]. The proposed DDN task can be considered as a combination of VON and VLN in terms of describing the target: the demand is described using natural *language* and the agent is asked to find *objects* that match the demand. The VLN task requires the agent to follow step-by-step instructions to navigate in a previously unseen environment. In contrast to the VLN task, the proposed DDN task provides high-level demand instructions and requires the agent to infer which objects satisfy the demand instructions within the current scene. The VON task can be broadly classified into two groups: closed-vocabulary object navigation and open-vocabulary (zero-shot) object navigation. In closed-vocabulary object navigation, the target object categories are pre-determined. For example, in the 2023 Habitat ObjectNav Challenge [45], there are 6 target object categories. These categories are usually represented as one-hot vectors. Therefore, many studies establish semantic maps [3, 10] or scene graphs [13] to solve the closed-vocabulary VON task. In open-vocabulary object navigation, the range of object categories is unknown and is usually given in the form of word vectors [11, 5, 12]. Many studies [11, 5] use pre-trained language models to obtain word embedding vectors to achieve generalisation in navigating unknown objects. Additionally, CLIP on Wheels [5] uses the CLIP model for object recognition. Different from the VON task, our proposed DDN task does not focus on the category of objects, but chooses "human demand" as the task context.

### 2.2 Large Language Models in Robotics

Recently, large language models (LLMs) [46, 23, 47–49] have attracted attention for their performance on various language tasks (e.g., text classification and commonsense reasoning). LLMs can exhibit a human level of common sense knowledge and can even be on par with human performance in some specialised domains. There is an increasing amount of research [37, 50–54] seeking to use the knowledge in LLMs to control or assist robots in performing a range of tasks. LM-Nav [37] accomplishes outdoor navigation without any training by combining multiple large pre-trained models (GPT-3 [23], CLIP [24] and ViNG [55]). In SayCan [51], the authors use a LLM to interpret high-level human instructions to obtain detailed low-level instructions, and then use pre-trained low-level instruction skills to enable the robot to complete tasks. However, SayCan assumes that the agent can access the object positions in navigation and only conduct experiments on one scene. PaLM-E [52] goes a step further by projecting visual images into the same semantic space as language, enabling the robot to perceive the world visually, but PaLM-E exhibits control that is either limited to table-level tasks or scene-level tasks with a map. Different from these approaches, the proposed method does not use LLMs for providing reasoning about instructions directly, but instead uses LLMs to learn attribute features about the objects. Such attribute features can help the agent learn an effective navigation policy in mapless scenes. This paper also uses LLMs to generate the required dataset.

## 3 Problem Statement

In the DDN task, an agent is randomly initialised at a starting position and orientation in a mapless and unseen environment. The agent is required to find an object that meets a demand instruction described in natural language (*e.g.*, "I am thirsty"). Formally, let $\mathcal{D}$ denote a set of demand instructions, $\mathcal{S}$ denote a set of navigable scenes, and $\mathcal{O}$ denote a set of object categories that exist in the real world. To determine whether a found object satisfies the demand, let $\mathcal{G} : D \times O \to \{0, 1\}$ denote a discriminator, outputting 1 if the input object satisfies the demand, or 0 if it does not. In the real world, $\mathcal{G}$ is effectuated by the user, but to quantitatively evaluate the proposed method and the baselines, $\mathcal{G}$ is effectuated by the DDN dataset collected in Sec. 4.

At the beginning of each episode, the agent is initialised at a scene $s \in \mathcal{S}$ with a initial pose $p_0$ (*i.e.*, position and orientation), and provided a natural language demand instruction $d =< d_1, d_2 \cdots d_L >\in \mathcal{D}$, where $L$ is the length of the instruction and $d_i$ is a single word token. In the demand instruction $d$, the task is made more challenging by not providing specific object names. This is to ensure the agent learns to reason in scenarios where these objects are not specified by the user. Then the agent is re-

quired to find an object $o \in \mathcal{O}$ to satisfy the demand instruction $d$ and only take RGB images as sensor inputs. The agent's action space is $\mathrm{MoveAhead}, \mathrm{RotateRight}, \mathrm{RotateLeft}, \mathrm{LookUp}, \mathrm{LookDown}$, and $\mathrm{Done}$. When the agent selects the action $\mathrm{Done}$, the agent is also required to output a bounding box $b$ to indicate the object that satisfies the demand within the current field of view RGB image. Then it is determined whether the success criteria have been satisfied. The success criteria for the DDN task include two conditions. 1) The navigation success criterion: this requires there is an object in the field of view that satisfies the demand instruction and the horizontal distance between the agent and the object is less than a threshold $c_{navi}$. 2) The selection success criterion: this requires that given that the navigation is successful, the intersection over union (IoU) [56] between the output bounding box $b$ and the ground truth bounding box of an object that satisfies the demand instruction is greater than a threshold $c_{sele}$. There is a 100 step limit to succeed during the episode.

## 4 Demand-Driven Navigation Dataset

Although mappings between objects and instructions will vary with the environment, a fixed mapping between demand instructions and objects is required to check selection success for training the agent in a specific environment, referred to as *world-grounding mappings* (WG mappings). The modifier "world-grounding" emphasises the connection to the specific environment of the real world and will vary with the environment. For example, the ProcThor dataset [25] and the Replica dataset [57] will have different WG mappings. To simulate real-world scenarios, it is assumed, unless otherwise stated, that the agent does not have access to the metadata of the object categories in the environment or the WG mappings that are used to determine selection success during training, validation and testing. The set of world-grounding mappings constructed in this paper form the demand-driven navigation dataset.

To construct the WG mappings, the metadata regarding object categories in the environment is obtained and GPT-3 is used to establish a fixed set of WG mappings between demand instructions and objects $\mathcal{F} : \mathcal{D} \rightarrow \mathcal{O}_f$, where $\mathcal{O}_f$ is a subset of $\mathcal{O}$. For example, $\mathcal{F}(\text{``I am thirsty''}) = \{\mathrm{Water}, \mathrm{Tea}, \mathrm{AppleJuice}\}$. Concretely, prompt engineering is used to inform GPT-3 of the object categories that may be present in the experimental environment. Then GPT-3 is used to determine which demands these objects can satisfy and returns this information in the form of a demand instruction and a statement of which objects present satisfy this demand instruction. It has been found that the WG mappings generated by GPT-3 are not absolutely accurate. Due to the presence of some errors in the generation, manual filtering and supplementation is then used to correct and enhance the dataset. These WG mappings $\mathcal{F}$ are used during training and testing, only to discriminate the selection success (*i.e.*, to effectuate discriminator $\mathcal{G}$). In total, approximately 2600 WG mappings are generated. Please see supplementary material for details of the generation process, the prompts, and the statistical features of the dataset.

## 5 Demand-Driven Navigation Method

It is important to note that if an object can fulfill a demand, it is because one of its functions or attributes can satisfy that demand. In other words, if multiple objects can fulfill the same demand, the objects should have similar attributes. For example, if your demand is to heat food, the corresponding objects that can satisfy the demand are a stove, an oven or a microwave, which all have similar attributes such as heating and temperature control. The relationship between attributes and instructions is not specific to a particular environment, but instead is universal common sense knowledge. Thus, this common sense knowledge can be extracted from LLMs. This universality turns the multi-object-target search into single-attribute-target search: the similarity between attributes reduces the complexity of policy learning. Therefore, we expect to extract the attribute features of the objects conditioned on the given demand from the LLM, *i.e.,*, demand-conditioned textual attribute features.

In this section, how the textual attribute features are learned from LLMs (Sec. 5.1), how they are aligned from text to vision (Sec. 5.2), and how to train the model (Sec. 5.3).

### 5.1 Textual Attribute Feature Learning

**Knowledge Extraction**    Given that the relationship between attributes and demand instructions is common sense knowledge, many mappings between demand instructions and objects are established

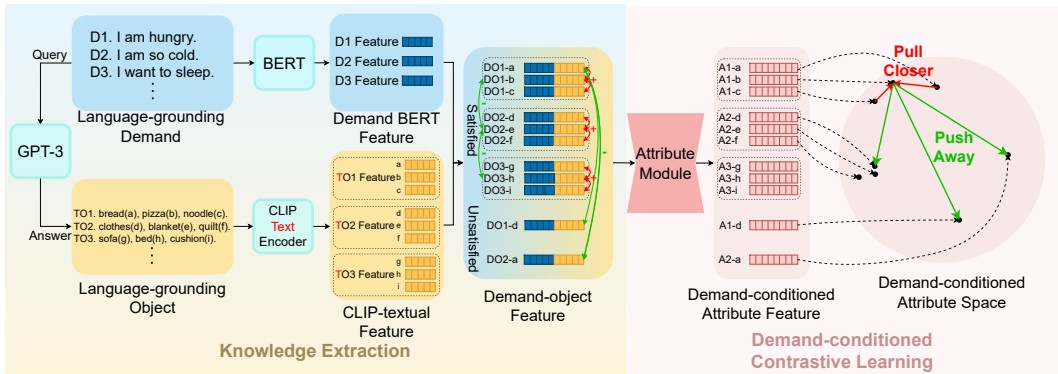

Figure 2: **Textual Attribute Feature Learning.** This diagram illustrates the process for textual attribute feature learning, where BERT and CLIP features are combined via concatenation and the attribute module is trained using contrastive learning. Green arrows and minus signs represent negative sample pairs. Red arrows and plus signs represent positive sample pairs. Models in this blue represent the use of pre-trained weights with frozen parameters.

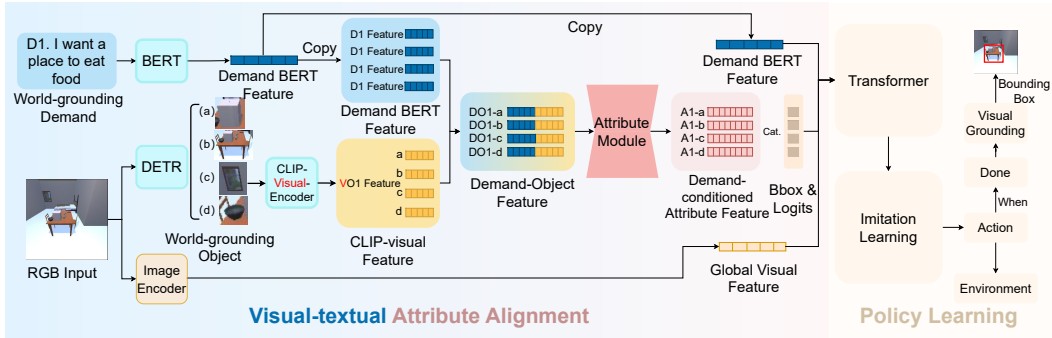

Figure 3: **Policy Learning with Visual Attribute Features.** This diagram illustrates the process for policy learning using visual attribute features, where BERT and CLIP features are combined via concatenation. The demand-conditioned attribute features obtained from the Attribute Module are fed into a transformer model together with BERT features and global image features. Models in blue represent the use of pre-trained models with frozen parameters.

with GPT-3, for learning demand-conditioned textual attribute features, referred to as *language-grounding mappings* (LG mappings). The modifier "language-grounding" emphasises that these mappings can be obtained from LLMs without the need for any connection to real-world situations. As LG mappings are independent of the real world and the experimental environment, they are suitable to be used in training the model. Concretely, as shown in Fig. 2, GPT-3 is used to generate a multitude of demand instructions in advance, called language-grounding demands (LG demands), and then GPT-3 is queried as to which objects can satisfy these demands. GPT-3 generates the resulting objects called language-grounding objects (LG objects). The LG demands and the LG objects compose the LG mappings. The BERT model is used to encode the LG demand instructions and obtain demand BERT features. The CLIP-Text-Encoder is used to encode LG objects and obtain CLIP-textual features. Then the demand BERT features and CLIP-textual features are concatenated together to obtain demand-object features (these features are named DO{X}-{Y}, with X representing the demand label and Y representing the object label).

**Demand-conditioned Contrastive Learning** Due to the fact that objects which satisfy the same demand show similar attributes and vice versa, contrastive learning is suitable to be used to train the Attribute Module. The way positive and negative samples in contrast learning are defined is next illustrated using the object and demand satisfaction relationships in Fig. 2: object a, b and c satisfy demand D1, object d, e and f satisfy demand D2, and object g, h and i satisfy demand D3. If two objects can satisfy the same demand, then the two demand-object features corresponding to these

two objects and the demand are positive sample pairs. For example, DO1-a and DO1-b is a positive sample pair. There are three categories of negative sample pairs. 1) For the same demand, if one object can satisfy it and the other cannot, the corresponding Dem-Obj features are negative sample pairs. For example, DO1-a and DO1-d is a negative sample pair. 2) For the same object, if it satisfies one demand but not the other, then the corresponding demand-object features are also negative sample pairs. For example, DO1-a and DO2-a is a negative sample pair. 3) The demand-object features with different objects and demands are also negative samples pairs. For example, DO1-a and DO2-c is a negative sample pair. Then demand-object features are used as the input to the Attribute Module to obtain the demand-conditioned attribute features (named A{X}-{Y}, with X representing the demand label and Y representing the object label). The demand-conditioned attribute features corresponding to positive sample pairs should be as close as possible in the demand-conditioned attribute feature space, and negative sample pairs should be as far away as possible. Therefore InfoNCE Loss [58] is used to train the Attribute Module. We use a 6-layer Transformer Encoder [59] to implement the Attribute Module.

## 5.2 Textual-Visual Alignment via CLIP

In Sec. 5.1, the demand-conditioned textual attribute features are learned using language-grounding mappings. However, the demand-conditioned textual attribute features are still at the language-level. To address this problem, the CLIP-Semantic-Space is used as a shared space for vision and text to allow the Attribute Module to obtain scene grounding information. The DETR [60] model is first used to segment object patches (referred to as world-grounding objects, WG object) in the field of view during navigation, and then the CLIP-Image-Encoder is used to project these object patches into the CLIP-Semantic-Space to obtain CLIP-Visual features. Both CLIP-visual features and CLIP-textual features are projected to the same CLIP-Semantic-Space, and thus they are aligned. The Demand BERT feature and CLIP-Visual features are concatenated (*i.e.*, demand-object features) and used as inputs to the Attribute Module trained in Sec 5.1. Note that the CLIP-Visual features come from the scene-grounding visual input. Therefore, the demand-conditioned attribute features obtained during navigation involve scene grounding information.

To summarise, in Sec. 5.1, contrast learning and language-grounding mappings generated by GPT-3 are used to allow the Attribute Module to learn to extract demand-conditioned attribute features of objects, which contain **common sense knowledge and human preferences**; in Sec. 5.2, CLIP's shared visual-textual semantic space is used to enable the Attribute Module to receive **scene-grounding information** during navigation.

## 5.3 Policy Learning and Visual Grounding Model

The features that are fed to a Transformer [59] are processed in a similar way to VTN [6]. Demand-conditioned features are concatenated with a bounding box and logits and input into a Transformer Encoder, then passed into a Transformer Decoder with Demand BERT features and global visual features (encoded by a pre-trained Vision Transformer [61, 62]). Imitation learning is used to train the proposed model. Approximately 27,000 trajectories generated by the $A^*$ algorithm [63] are collected and used for training.

The Visual Grounding Model (VG model) is essentially a classifier built with a Transformer Encoder. The DETR model is used to segment the object instances on the current image, and then the features of the last layer of DETR for the highest $k$ patches of DETR logits, the global image features encoded by RESNET-18 [64], the Demand BERT features and a CLS token, are combined and fed to the VG model, and then the output features corresponding to the CLS token are used for classification. The VG model is used for all the baselines that perform DDN and the proposed method.

For more details about policy learning, the fine-tuning of the image encoder and DETR, the pipeline and training procedure of the demand-based visual-grounding model, and hyperparameters, please see the supplementary material.

Table 1: **Quantitative comparison over baselines and ablations.** NSR: navigation success rate. NSPL: navigation success rate weighted by the path length. SSR: selection success rate. $^*$ indicates that the algorithm leverages scene metadata. The parentheses show the sample standard deviation.

| | Seen Scene | | | | | | Unseen Scene | | | | | |
| | Seen Ins. | | | Unseen Ins. | | | Seen Ins. | | | Unseen Ins. | | |
| Method | NSR | NSPL | SSR | NSR | NSPL | SSR | NSR | NSPL | SSR | NSR | NSPL | SSR |
|---|---|---|---|---|---|---|---|---|---|---|---|---|
| Random | 5.2(0.72) | 2.6(0.66) | 3.0(0.93) | 3.7(0.76) | 2.6(0.24) | 2.3(0.63) | 4.8(0.48) | 3.3(0.41) | 2.8(0.8) | 3.5(0.18) | 1.9(0.15) | 1.4(0.7) |
| VTN-demand | 6.3(1.7) | 4.2(1.3) | 3.2(1.3) | 5.2(0.83) | 3.1(0.75) | 2.8(0.91) | 5.0(1.0) | 3.2(1.1) | 2.8(1.2) | 6.6(0.9) | 4.0(1.6) | 3.3(1.0) |
| VTN-CLIP-demand | 12.0(3.0) | 5.1(1.5) | 5.7(2.3) | 10.7(4.3) | 3.5(0.5) | 5.0(1.0) | 10.0(3.0) | 3.6(1.3) | 4.0(3.0) | 9.3(5.3) | 3.9(0.1) | 4.0(3.0) |
| VTN-GPT* | 1.6(1.1) | 0.5(0.70) | 0(0) | 1.4(0.65) | 0.4(0.57) | 0.5(0.35) | 1.3(0.76) | 0.2(0.20) | 0.3(0.44) | 0.9(0.20) | 0.4(0.34) | 0.5(0.35) |
| ZSON-demand | 4.2(1.2) | 2.7(0.78) | 1.9(1.1) | 4.6(2.0) | 3.1(1.3) | 2.0(0.6) | 4.1(0.6) | 2.9(0.2) | 1.2(0.4) | 3.5(0.6) | 2.4(0.7) | 1.1(0.6) |
| ZSON-GPT | 4.0(2.5) | 1.1(2.4) | 0.3(0.27) | 3.6(2.6) | 1.9(2.6) | 0.3(0.27) | 2.5(1.2) | 0.7(0.4) | 0.2(0.27) | 3.2(1.2) | 0.9(0.1) | 0.2(0.27) |
| CLIP-Nav-MiniGPT-4 | 4.0 | 4.0 | 2.0 | 3.0 | 3.0 | 2.0 | 4.0 | 3.7 | 2.0 | 5.0 | 5.0 | 3.0 |
| CLIP-Nav-GPT* | 5.0 | 5.0 | 4.0 | 6.0 | 5.5 | 5.0 | 5.5 | 5.3 | 4.0 | 4.0 | 3.0 | 2.0 |
| FBE-MiniGPT-4 | 3.5 | 3.0 | 2.2 | 3.5 | 3.2 | 2.0 | 3.5 | 3.5 | 2.0 | 4.0 | 4.0 | 3.5 |
| FBE-GPT* | 5 | 4.3 | 4.3 | 5.5 | 5.0 | 5.5 | 4.5 | 4.3 | 4.5 | 5.5 | 5.0 | 5.5 |
| GPT-3-Prompt* | 0.3 | 0.01 | 0 | 0.3 | 0.01 | 0 | 0.3 | 0.01 | 0 | 0.3 | 0.01 | 0 |
| MiniGPT-4 | 2.9 | 2.0 | 2.5 | 2.9 | 2.0 | 2.5 | 2.9 | 2.0 | 2.5 | 2.9 | 2.0 | 2.5 |
| Ours_w/o_attr_transformer | 15.6(10.3) | 6.9(1.9) | 6.0(1.0) | 15.1(3.6) | 8.9(2.2) | **7.2(2.3)** | 12.3(0.3) | 4.5(2.3) | 4.7(2.3) | 11.7(7.3) | 5.4(6.8) | 2.7(0.3) |
| Ours_w/o_attr_pretrain | 13.1(3.5) | 5.6(1.6) | 4.8(2.0) | 13.8(2.8) | 6.2(1.8) | 5.3(2.3) | 12.7(1.5) | 5.7(0.98) | **6.3(2.5)** | 11.8(0.58) | 5.8(0.57) | 3.6(0.29) |
| Ours_w/o_BERT | 12.2(12.2) | 4.2(3.5) | 5.8(8.7) | 8.4(9.3) | 2.8(2.3) | 4.0(3.5) | 7.8(10.7) | 2.5(2.3) | 2.6(0.8) | 8.4(14.3) | 2.7(2.4) | 3.8(1.7) |
| Ours | **21.5(3.0)** | **9.8(1.1)** | **7.5(1.3)** | **19.3(3.3)** | **9.4(1.8)** | 4.5(1.8) | **14.2(2.7)** | **6.4(1.0)** | 5.7(1.8) | **16.1(1.5)** | **8.4(1.1)** | **6.0(1.2)** |

# 6 Experiments

## 6.1 Experimental Environment

The AI2Thor simulator and the ProcThor dataset [25] are used to conduct the experiments. Throughout the experiments, 200 scenes in each of ProcThor's train/validation/test sets are selected for a total of 600 scenes. There are 109 categories of objects that can be used in the environments to satisfy the demand instructions. 200 WG mappings are selected for training and 300 for testing from our collected DDN dataset. The WG mappings used for validation are the same as those used for training. Our experiments can all be done on 2×A100 40G, 128-core CPU, 192G RAM. To complete training for a main experiment takes 7 days.

## 6.2 Evaluation Method and Metrics

A closed-vocabulary object navigation algorithm, VTN [6], and an open-vocabulary object navigation algorithm, ZSON [11], are chosen as the original VON baselines. To fit these baselines to the proposed DDN task, several modifications are made to the algorithms as follows and the results are shown in the form of a postfix shown in Tab. 5.3. The VTN-CLIP means that we replace the DETR features used in VTN with CLIP features. The "demand" suffix means the original one-hot vectors (closed-vocabulary) or word vectors (open-vocabulary) are replaced with the demand BERT features. The "GPT" suffix means that the GPT-generated language-grounding objects are used as the parsing of the demand instruction, and then the parsing results are used as the input for object navigation. These VON baselines and their variants should demonstrate the difficulty of the DDN task and the fact that DDN tasks are not easily solved by utilising language models for reasoning. Finally, five variants of the VON algorithm are introduced: *VTN-CLIP-demand*, *VTN-demand*, *VTN-GPT*, *ZSON-demand*, *ZSON-GPT*.

The popular large language model GPT-3 and the open source multi-modal large language model MiniGPT-4 are also used for navigation policies and recognition policies, resulting in two baselines: *MiniGPT-4* and *GPT-3-Prompt\**. GPT-3 and MiniGPT-4's test results, should provide insight into the strengths and weaknesses of large language models for scene-level tasks. Random (*i.e.*, randomly select an action in the action space) is also used as a baseline for showing the difficulty of the DDN task. We also employ a visual-language navigation algorithm, CLIP-Nav [65] and a heuristic exploration algorithm, FBE [66] as navigation policies, with GPT-3 and MiniGPT-4 serving as recognition policies, resulting in four baselines: *CLIP-Nav-GPT*, *CLIP-Nav-MiniGPT-4*, *FBE-GPT*, and *FBE-MiniGPT-4*.

The VTN-CLIP-demand, VTN-GPT, and VTN-demand algorithms are trained using the same collected trajectory as our method. $^*$ indicates that the algorithm leverages environmental metadata. Navigation success rate, navigation SPL [67], and selection success rate are used as metrics for comparison. For more details on metrics and baselines (about the metadata usage of the algorithms, prompts used in GPT-3 and MiniGPT-4, etc), please see the supplementary materials.

## 6.3 Baseline Comparison

Detailed and thorough experiments are conducted to demonstrate the difficulty of the DDN task for the baselines and the superiority of the proposed method, shown in Tab. 5.3.

The results of *VTN-demand* are slightly better than *Random*, indicating that *VTN-demand* has learned to some extent the ability to infer multiple potential target objects simultaneously. However, *ZSON-demand* has lower performance than *Random*. Considering that *ZSON-demand* uses CLIP for visual and text alignment, this lower result indicates that CLIP does not perform well on alignment between instructions and objects. *VTN-GPT* and *ZSON-GPT* demonstrate very poor performance. This is likely to be due to the fact that the language-grounding objects given by GPT-3 have a high likelihood not to be present in the current environment, which leads to a meaningless search by the agent. *VTN-GPT* and *ZSON-GPT* also show that converting DDN to VON through a pre-defined mapping between instructions to objects is not effective. The superior performance of *VTN-CLIP-demand* compared to *VTN-demand* underscores the contribution of CLIP in effectively extracting features of objects. The two variants about *CLIP-Nav* do not perform well, and we argue that this may be attributed to a significant disparity in content between the step-by-step instructions in VLN and the demand instructions in DDN. Since ProcThor is a large scene consisting of multiple rooms, the results show that heuristic search *FBE* is not efficient.

The performance of *GPT-3+Prompt* is significantly lower than *Random*. This lack of performance is due to the lack of visual input to the *GPT-3+Prompt* method and relying only on manual image descriptions resulting in insufficient perception of the scene. Moreover, during the experiment, it is observed that *GPT-3+Prompt* rarely performs the Done action, and approximately 80% of the failures are due to exceeding the step limit, suggesting that *GPT-3+Prompt* lacks the ability to explore the environment which leads to its inability to discover objects that can satisfy the given demand. With visual inputs, *MiniGPT-4* outperforms *GPT-3+Prompt* despite its relatively small model size, indicating that visual perception (*i.e.*, scene grounding information) is of high significance in scene-level tasks. *MiniGPT-4* still does not perform as well as *Random*. By observing *MiniGPT-4*'s trajectories, two characteristics are noticed: 1) the agent tends to turn its body left and right in the original place and turn the camera up and down, and does not often move; 2) the selection success rate is very close to the navigation success rate. It is therefore considered that *MiniGPT-4*'s task strategy is as follows: observe the scene in place and then choose to perform the Done action if the agent identifies objects in sight that can satisfy the demand, and output the selected object. By the fact that the selection success and navigation success rates are close, it can be found that *MiniGPT-4* satisfactorily recognises the semantics of objects and infers whether they meet the user's demand. However, the low navigation success rate implies that *MiniGPT-4* cannot determine whether the distance between it and the object is less than the threshold value $c_{navi}$.

## 6.4 Ablation Study

We conduct three ablation experiments, with specific details as follows:

- Ours w/o attr pretrain: we use the transformer network in the Attribute Module without demand-conditioned contrastive learning to pretain it.
- Ours w/o attr transformer: we replace the transformer network with a MLP layer in the Attribute Module.
- Ours w/o BERT: we replace the BERT endocer with a MLP layer.

*Ours* surpasses all baselines. The Attribute Module turns multi-object-target goal search into single-attribute-target search, which makes the agent no longer need to find potentially different objects in the face of the same demand instruction, but consistent attribute goals. The consistency between demands and attributes reduces the policy learning difficulty. In *Ours w/o BERT*, we observe a significant performance drop, especially in the setting with unseen instructions. This finding indicates that the features extracted by BERT are highly beneficial for generalisation on instructions. *Ours w/o attr pretrain* shows that our proposed demand-conditioned contrastive learning has a significant improvement to performance. After demand-conditioned contrastive learning, the Attribute Module extracts generalised attribute features from LLMs and LLMs provide sufficient common sense knowledge and interpretation of human preferences for the Attribute Module to learn. Once the common sense knowledge extracted from the LLMs is lost, the Attribute Module can only learn

the association between attributes and instructions from the sparse navigation signals of imitation learning. However, *Ours w/o attr pretrain* still surpasses all baselines (including VTN with a similar model structure), which is probably due to the fact that the object semantic information [68] contained in the CLIP-visual features helps the Attribute Module to understand the demand instructions and learn the relationship between objects and instructions to some extent through imitation learning. The performance decrease observed in *Ours w/o attr transformer* compared to *Ours* suggests that the transformer network is more effective than MLP in capturing the attribute features of objects.

## 7    Conclusion and Discussion

This paper proposes Demand-driven Navigation which requires an agent to locate an object within the current environment to satisfy a user's demand. The DDN task presents new challenges for embodied agents, which rely heavily on reasoning with common sense knowledge, human preferences and scene grounding information. The existing VON methods are modified to fit to the proposed DDN task and they have not been found to have satisfactory performance on the DDN task. The LLM GPT-3 and the open source MM-LLM, MiniGPT-4 also show some drawbacks on the DDN task, such as being unable to determine the distance between the object and the agent from visual inputs. This paper proposes to learn demand-conditioned object attribute features from LLMs and align them to visual navigation via CLIP. Such attribute features not only contain common sense knowledge and human preferences from LLMs, but also obtain the scene grounding information obtained from CLIP. Moreover, the attribute features are consistent across objects (as long as they satisfy the same demand), reducing the complexity of policy learning. Experimental results on AI2thor with the ProcThor dataset demonstrate that the proposed method is effective and outperforms the baselines.

**Limitations and Broader Societal Impacts**    As we do not have access to GPT-4 with visual inputs and PaLM-E, we can only use the open source MM-LLM (*i.e.*, MiniGPT-4) to test on the proposed DDN task, so we only have the possibility to conjecture the strengths and weaknesses of MM-LLMs on our task from the performance of MiniGPT-4. We expect that our work will advance the performance of MM-LLMs on scenario-level tasks. The proposed DDN task is in the field of visual navigation for finding objects that satisfy human demands. To the best of our knowledge, there are no observable adverse effects on society.

## Acknowledgments and Disclosure of Funding

This work was supported by National Natural Science Foundation of China - General Program (62376006) and The National Youth Talent Support Program (8200800081). We also thank Benjamin Redhead for correcting and polishing the grammar and wording of our paper.

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
