# OpenReview forum: "Find What You Want: Learning Demand-conditioned Object Attribute Space for Demand-driven Navigation"
_NeurIPS.cc/2023/Conference — NeurIPS 2023 poster_

### Official Review · Reviewer_FyRT · 2023-07-05

**Soundness:** 3 good
**Presentation:** 3 good
**Contribution:** 3 good
**Rating:** 7
**Confidence:** 4

**Summary:**

This paper proposes Demand-driven Navigation (DDN), which leverages the user’s demand as the task instruction and prompts the agent to find an object which matches the specified demand. This paper also proposes a method of first acquiring textual attribute features of objects by extracting common sense knowledge from a large language model. These textual attribute features are subsequently aligned with visual attribute features using Contrastive Language-Image Pre-training (CLIP). They experimented on AI2Thor with the ProcThor dataset and demonstrated that the visual attribute features improve the agent’s navigation performance and outperform the baseline methods.

**Strengths:**

The paper flows nicely with good motivation of the novel task and its challenges clearly stated.
The proposed method makes sense and comparison is good.

**Weaknesses:**

1.	L40, the second condition is a bit questionable. I don’t think it is a necessary requirement of only searching objects that are in the scene. It is probably the case in current benchmark setup, however it is not a constraint for the research task. Subsequently, it might also impact the definition of “navigation failure” at L48. Asking the robot to search for a non-existent object in the scene and the robot could not find it. This should not be defined as a failure from my point of view. Rather, if you ask a robot to look for some non-existent object and it reports a finding, this should be counted as a failure as it is a clear false positive detection. Nevertheless, the new task DDN does not necessarily remove ‘requirement’ 2) as you might anyway asks for a demand that no object in the scene can satisfy.
2.	From L171, my impression is that the problem is constrained by only one demand d at a time. In reality, one could have multiple demands to further constrain what object they might want to search. For example, something “quench thirst” and “contains caffeine”.
3.	It will be more convincing with more benchmark datasets. In addition to AI2Thor, there is Habitat Challenge on ObjectNav which can be exploited, plus it is based on 3D scan of real scenes.

**Questions:**

1) Can we handle a search composed with multiple demands, with the current method?
2) Why other benchmark dataset are not used for evaluation? is there any justification?

**Limitations:**

The authors discussed both limitations and the societal impact.

---

> ### Author Rebuttal · Authors · 2023-08-08
>
> REPLY: Thank you! We have clarified some questions, address your concerns, and hope to hear back from you if you have further questions!
>
> **Q1**: L40, the second condition is a bit questionable.
>
> A1: Thank you very much for your insightful suggestion. We understand the concern about the second condition in L40. In the current VON task, the robot is typically asked to find an object that is guaranteed to exist in the current scene. However, considering real-world applications, users may request the robot to find an object that they are unsure exists in the scene. In terms of satisfying the user, the robot does fail to find it, which is considered a failure. But we also agree with you that it's not really a navigation failure because the object doesn't exist in the scene; we'll revise the statement here to say  "failure to satisfy the user's demand" rather than "navigation failure".
>
> **Q2** : From L171, my impression is that the problem is constrained by only one demand d at a time. Can we handle a search composed with multiple demands, with the current method?
>
> A2: Thank you very much for your valuable suggestion. What you are describing is a more refined instruction, and we have some similar examples in our dataset, such as "I need a place to rest" and "I need a soft place to rest". The former can be "bed, wooden chair, sofa" while the latter cannot be "wooden chair" but "bed, sofa". The latter contains two demands in its instruction: "soft" and "available for rest".
>  In future work, we will add some explicit personal preferences as well as more complex demand instructions, including negation of object samples, combining multiple demands, and prioritising between objects. Theoretically our method works for any demand expression including multiple demands, since any demand corresponds to some attributes that match the demand.
>
> **Q3**:It will be more convincing with more benchmark datasets.
>
> A3: Thank you very much for your reminder. Our dataset generation process and navigation methods can be easily migrated to any other dataset, including the scene dataset used in the Habitat Challenge. We will later generate some DDN dataset (e.g., Matterport3D) for the current mainstream scene datasets for training and testing.

---

> > ### Comment · Reviewer_FyRT · 2023-08-16
> > **Thanks for the reply**
> >
> > I thank authors for the response, and I would sincerely suggest to add these clarifications in the paper. As I mentioned in the original comments, the new task makes sense and the method is convincing with the supported comparison. I'd appreciate a good clarity in the statement in terms of the assumptions and limitations.

---

> > > ### Author Response · Authors · 2023-08-16
> > >
> > > Thank you for your valuable suggestion! We will add these clarifications about assumptions and limitations in the main paper.
> > >
> > > Many thanks again for your response! We hope to hear back if you have further questions!

---

### Official Review · Reviewer_i2zg · 2023-07-05

**Soundness:** 3 good
**Presentation:** 2 fair
**Contribution:** 2 fair
**Rating:** 5
**Confidence:** 5

**Summary:**

This paper presents a new visual navigation setting, where the goal is not specified by objects or images but described by a sentence. The sentence encodes the essential information to search for specific objects during navigation. Different from VLN, the task is able to analyze the demand within each sentence rather than step-by-step language guidance. Common sense and knowledge will be explicitly extracted from a large language model. To align the visual and attribute features, CLIP is employed. Overall, the motivation of this work is reasonable and interesting.

# Post rebuttal
The authors have addressed most of my concerns. If the authors can provide some visualization results, that would be more convincing. Therefore I changed my score to accept.

**Strengths:**

The motivation of this work is interesting. Providing a demand description to an agent would enable the agent to search for not only one object in order to complete the specified goals.

The authors also provide semi-automatically generated data for this new task. This would be complementary to the existing VLN or VN tasks.

**Weaknesses:**

The introduction part is overly lengthy. The authors exert three pages to describe the motivations of this work, making reading quite tedious. I highly suggest the authors could trim the introduction part a bit.

The natural questions come to this task is whether the proposed method can complete object-goal navigation after training? For example, after the network is trained based on demand driven sentences, whether it can be used as a object-goal navigation agent?

The common knowledge or sense is pre-defined. In the illustrative figures, it seems a demand may correspond to three different objects. Whether this would restrict the options?

In L189, the WG mappings are different depending on the environment. I am not sure whether this implies that these WG mappings need to be specified manually. If so, this may contradict with the original motivation of this work, where humans may not know the environment in advance.

**Questions:**

My questions mainly focus on two tasks:

In this comparisons, the results of some baseline methods are significantly lower than the results reported by their original papers. Therefore, I am wondering how the authors adapt their methods to this setting?

A few works leverage CLIP for VLN or VN tasks.
CLIP-Nav: Using CLIP for Zero-Shot Vision-and-Language Navigation
Vision-and-Language Navigation: Interpreting visually-grounded navigation instructions in real environments
These works can be adopted for comparison.

As the searched objects have strong association with the demand, what if some demands cannot be processed properly? For example, I am thirsty but I cannot drink cold. Simply providing demands and their corresponding objects would lead to overfitting.

**Limitations:**

There is no negative societal impact.

---

> ### Author Rebuttal · Authors · 2023-08-08
>
> REPLY: Thank you! We have clarified some questions, address your concerns, and hope to hear back from you if you have further questions!
>
> **Q1**: The introduction part is overly lengthy.
>
> A1: Thank you for your valuable feedback. We understand your concern, and we agree that the introduction may have become lengthy due to the detailed descriptions of the new task (DDN) and the comparison with the old task (VON). To streamline the introduction, we will remove redundancies and avoid duplicating information that will be discussed in detail in later sections. Additionally, we will consider moving some of the descriptions of the DDN task to a separate section to maintain a better flow in the paper.
>
> **Q2**: The natural questions come to this task is whether the proposed method can complete object-goal navigation after training?
>
> A2: Thank you for raising this question. Theoretically, our proposed method can be utilized for object-goal navigation, by translating the target object into a demand instruction.  We use the template "I want a $object$" to transform an object name into the form of a demand instruction, and then test our trained DDN model in the Object Navigation setting (consistent with the setting of the -object suffix in our paper). The results of the experiment are 9.0/9.0 (SR/SPL) in seen scene and 8.5/8.3 (SR/SPL) in unseen scene. Compared to VTN-object and ZSON-object, the results demonstrate that our method also performs well on object navigation.
>
>
> **Q3** : The common knowledge or sense is pre-defined. In the illustrative figures, it seems a demand may correspond to three different objects. Whether this would restrict the options?
>
> A3: Thank you so much for pointing that out. We apologize for any confusion. To clarify, common knowledge or sense is not pre-defined; rather, it arises from human consensus and understanding of the world. In the illustrative figures, we provided an example explanation of the DDN task and our method, showing that a demand may correspond to three different objects. However, this example is not intended to limit the number of objects that can satisfy a demand. In real-life scenarios, a demand can indeed be fulfilled by numerous objects, and the number of possible options can vary significantly depending on the environments.To demonstrate the flexibility and diversity of the DDN task, we use GPT-3 to generate 10 object categories for each demand instruction in a DDN dataset, showcasing the multiple possibilities that can exist for a single demand (please refer to the description of LG mapping in the Supplementary Material 8.2.1).
>
> **Q4**: If so, this may contradict with the original motivation of this work, where humans may not know the environment in advance.
>
> A4: Thanks for pointing that out. During the training process in a simulator, we require an "expert" to assess whether the object found by the agent satisfies the demand or not, which allows the "expert" to provide the agent with correct training signals. The WG mapping serves as this "expert" during training.
> Since different environments can have distinct categories of objects, different WG mappings are needed for training in different environment. However, it is crucial to note that when the agent is fully trained and deployed in a real-world environment, it no longer requires WG mapping. Once a user gives a demand instruction, the agent will locate an object and present it to the user. In this case, the user does not need to know the details of the scene or the objects present; they only need to judge whether the object found by the agent satisfies their demand or not.
>
> **Q5**: Therefore, I am wondering how the authors adapt their methods to this setting?
>
> A5: Thank you for raising this question. To ensure a fair and comprehensive benchmark of the baseline methods, we employed different adaptation protocols for each one. For the -demand suffix baselines, we directly replaced the original VON input with the BERT feature of the demand instruction. The -GPT suffix baselines involve asking GPT-3 what objects satisfy a given demand instruction and then providing the answered object as input to the VON baselines. We have described the training and testing protocols for each baseline in detail in the Supplementary Material 8.3.2, providing a comprehensive explanation of how we adapted the VON method to suit the DDN task.
>
> **Q6**: VLN can be adopted for comparison.
>
> A6: Thank you for your insightful suggestion. We have taken your advice into account and included two additional baselines in our experiments. These baselines utilize CLIP-Nav as the navigation policy and GPT-3 and MiniGPT-4 as recognition policies, respectively. It is important to note that while the task instructions for vision-language navigation (VLN) are typically step-by-step, the instructions for the Demand-Driven Navigation (DDN) task revolve around the concept of "demand" for describing an object. As a result, we have adapted CLIP-Nav for the DDN task without utilizing its instruction breakdown. **Due to Rubuttal character limitations, the results of the experiment are shown in the attached PDF in Common Response**.
>
> **Q7** : As the searched objects have strong association with the demand, what if some demands cannot be processed properly?
>
> A7:  Thank you very much for your valuable suggestion. What you are describing is a more refined instruction, and we have some examples in our dataset, such as "I need a place to rest" and "I need a soft place to rest". The former can be "bed, wooden chair, sofa" while the latter cannot be "wooden chair" but "bed, sofa". In future work, we will add some explicit personal preferences as well as more complex demand instructions, including negation of object samples, combining multiple demands, and prioritising between objects.

---

> > ### Author Response · Authors · 2023-08-16
> > **More explanations about baselines' low performance**
> >
> > Thank you so much for reading the following. We explain the question in Q5 in more detail, especially in response to why the results of some baseline methods are significantly lower than the results reported by their original papers.
> >
> > **Q8**: In this comparisons, the results of some baseline methods are significantly lower than the results reported by their original papers.
> >
> > A8: There are several factors contributing to this observation.
> >
> > (1) It's important to highlight that the scope of object categories within the DDN task has expanded significantly, encompassing a total of 109 categories. In contrast, the original VON papers focused on a narrower range of objects: 22 categories in the case of VTN, and 6-21 categories for ZSON. This broader object category coverage inherently introduces greater complexity.
> >
> > (2) The DDN task involves the utilization of natural language instructions, resulting in a considerably wider  description space than the VON task. This expanded description space inherently escalates the level of task complexity.
> >
> > (3) the VON methods do not take into account the many-to-many object-instruction mapping phenomenon present in the DDN task, resulting in their lack of reasoning about the combination of instructions and scenes.
> >
> > These are the reason why the VON methods perform lower on the DDN task than reported in their original VON paper.
> >
> > Our method uses GPT-3 to generate numerous language-grounding mappings for learning demand-conditioned object attribute features. By leveraging CLIP's capability to align visual and textual information, our method integrates instruction and scene details.
> >
> >
> > We hope to hear back if you have further questions!

---

> ### Author Response · Authors · 2023-08-20
>
> Dear reviewer #i2zg
>
> Thank you very much for your time and effort spent in reviewing our paper. We appreciate your valuable suggestions for our papers.
> **As the discussion period is ending soon, we would like to  kindly request that you take into consideration the possibility of adjusting your score.** Please let us know whether you have further concerns. We are sincerely waiting for your response!
>
> Best wishes,
>
> Authors of 2423

---

### Official Review · Reviewer_7yMw · 2023-07-05

**Soundness:** 2 fair
**Presentation:** 3 good
**Contribution:** 3 good
**Rating:** 5
**Confidence:** 4

**Summary:**

This paper introduces a new task called Demand-Driven Navigation (DDN) that, unlike previous Visual Object Navigation (VON) tasks that evaluate the ability of an agent to find a specific object in an unknown environment, considers fulfilling the demand of a human. This new task is motivated by the lack of real-world grounding of current VON tasks that either require an agent to find an instance of an object category from a pre-defined fixed vocabulary or a language-specified object in an open-vocabulary fashion. However, in a real environment, a specific object might not be present or, if thinking from the point of view of fulfilling a human’s demand, other objects might be equally as good as the targeted one. As a result, authors suggest querying an autonomous agent with a language demand instruction. This allows one to be more flexible with respect to environments but also requires common sense knowledge, an understanding of how objects can be used, and where they are likely to be located in a scene.

The paper evaluates several baselines inspired by previous literature to show the introduced task is hard and cannot be solved with currently known methods. A new approach to solving the DDN task is thus introduced in this paper. The main goal is to learn a mapping between human demands and attributes of objects that can fulfill them. At training time, a GPT-3 model is thus used to generate a series of demands and objects that can meet each of them. Each demand is encoded by a BERT model and each object is encoded by a CLIP text encoder. For each demand, a demand-object vector is created by concatenating the demand representation and an object representation. An attribute module takes the demand-object vector as input and is trained with contrastive learning to extract representations that are as close as possible for different pairs sharing the same demand. This attribute module is then used in the final policy that is composed of a Transformer model and is trained with imitation learning. The paper shows this new method outperforms other baselines.

**Strengths:**

1. The introduced task is very interesting: querying an agent with a human demand seems more aligned with what is needed when deploying a robot in a real environment to help humans. Authors have properly motivated the caveats of current VON tasks in the literature.
2. The extraction of object attributes seems very relevant and well-motivated. Learning attributes is indeed a way to guide the learning of common sense knowledge and experiments showcase the gain in performance it allows to reach.
3. Authors propose diverse baselines to evaluate the performance of current language models in robotics/navigation tasks.

**Weaknesses:**

1. [Major] The introduced method is composed of many different modules and pre-trained models (GPT-3, BERT, CLIP text encoder, CLIP vision encoder, Attribute Module, DETR, Image Encoder, Policy Transformer, Visual Grounding model). It is not clear what parts are the most important and whether the overall method could maintain the same performance without some of these building blocks. Additional ablation studies would be very interesting.
2. [Major] Most baselines except *Ours* showcase very low and close performance. However, section 6.3 discussing the experimental results is quite long and detailed. I am not convinced authors can draw as many conclusions as they do when we consider how close all baselines are in terms of average performance (even more when considering the standard deviation).
3. [Major] This remark is very related to the previous one. All concurrent baselines reach very low performance. Could it be that these methods were not trained to convergence or would simply require much more training? The authors mention 1.8M training frames in the paper, which seems rather small compared with the number of frames required to train baselines in other navigation tasks (generally closer to 100M training frames).
4. [Minor] When presenting baselines in section 6.2, a lot of information is missing. The paper refers to details given in the supplementary material. When reading this supplementary information, we can understand the introduced baselines. I would suggest including this information in the main paper directly.

**Questions:**

Four questions (1.-4.) were already asked in the "Weaknesses" section. I would like authors to address these concerns. I also add two additional questions (5., 6.), that do not appear to me as paper weaknesses but rather required clarifications:

1.-4. See "Weaknesses" section.

5. [Major] When describing the *Random* baseline, authors say it is about randomly selecting an action in the action space. I thus do not understand the difference between *Random-object* and *Random-demand*. Further clarifications are needed.
6. [Major] Performance for *GPT-3+Prompt** and *MiniGPT-4* in Table 1 is always the same independently from the scene (seen/unseen) and instructions (seen/unseen). This should either be corrected or explained.

**Limitations:**

The paper mentions limitations regarding the drawn conclusions about the performance of current language models on the DDN tasks. Authors explain they did not have access to the code for recent methods such as GPT-4 with visual inputs or PaLM-E, and thus were not able to evaluate these methods. It is a good thing to mention this limitation, but their experiments were already conducted with many recent models, which might still allow them to draw relevant conclusions (see “Weaknesses” section for remarks about the drawn conclusions, which are however orthogonal to the mentioned limitations in this section).

---

> ### Author Rebuttal · Authors · 2023-08-09
>
> REPLY: Thank you! We have clarified some questions, address your concerns, and hope to hear back from you if you have further questions!
>
> **Q1**: Additional ablation.
>
> A1: Thank you very much for your advice. We show ablation experiments for pre-training of attribute modules in the main paper. Now we additionally add two ablation experiments on BERT and Attribute Module's network: replacing BERT with MLP and AttributeTransformer with a 4-layer MLP, and the results of the experiments are shown **in the attched PDF file in Common Response**. The results reveal that BERT features are essential for learning a robust navigation policy, especially on unseen instructions. The ablation results on Attribute Module's network show that the transformer outperforms the MLP in learning attribute features.
>
> Since we need to rely on CLIP's ability to align text and vision, we cannot perform ablation experiments on CLIP. DETR, VG Model and GPT-3 are just tools we use to make the pipeline complete, they can be replaced by any model with similar functionality. For example, DETR can be replaced by Faster-RCNN, VG model can be replaced by SAM. We regret not being able to provide ablation experiments on the policy transformer and image encoder due to computational and time constraints during the rebuttal period. However, our network follows the structure of VTN, where you may find some related ablation experiments.
>
> **Q2**: I am not convinced authors can draw as many conclusions.
>
> A2: Thank you for bringing up this concern. Our conclusions were derived from a comprehensive analysis of the agent's trajectory and decision-making data. We acknowledge that we fell short in providing this crucial information and apologize for the oversight.
>
> To explain the claim in L334-335, the difference between ZSON-object and ZSON-demand needs to be explained. Both ZSON-object and ZSON-demand were trained using Image Navigation (task input is CLIP-vision-feature of the target image), but for testing, ZSON-object was tested with Object Navigation (with the task input being CLIP's text feature of the target object's name), while ZSON-demand was tested with the DDN task (with the task input being CLIP's text feature of the demand instruction). ZSON 's motivation is to rely on CLIP's ability to align between text and vision to accomplish zero-shot object navigation. This alignment ability is reflected in the cosine similarity of the features.The average cosine similarity between object image features and object name features in ZSON-Object is 0.28; whereas the average cosine similarity between object image features and instruction features in ZSON-demand is only 0.22. So we argue that the alignment between instructions and objects is not good.
>
> Our contention in L336-338 is rooted in rigorous statistical analysis. Our findings indicate a 54.4% probability that the object suggested by GPT-3 may not even exist within the current environment.
>
> The reason we claim in L343-345 is that we counted the distribution of GPT-3+Prompt's actions and the number of the episodes that exceed the step limit, and found that 66.41% of the steps were spent rotating in place or adjusting the camera, whereas these actions accounted for only 32.41% of the expert's data in our traject dataset; and also that 80% of its episodes of failure were due to exceeding the 100-step limit.
>
> Regarding MiniGPT-4, we statistically obtained that "MoveAhead" accounts for only 15% of the total number of actions, while rotating and adjusting the camera in place accounts for 66% ("MoveAhead" accounts for 66.41% and rotating and adjusting the camera in place accounts for 32.41% in the expert trajectories); and the average episode length of MiniGPT-4 is 4.38 (the average trajectory length of the expert trajectories is 27.24). This suggests that MiniGPT-4 does not tend to move around looking for objects, but rather observes in place and then quickly decides on the target object to end the episode.
>
> **Q3**: Could it be that these methods were not trained to convergence or would simply require much more training?
>
> A3: We set up the validation set for model selection (picking the model with the highest NSR on validation). We found that long before the 1.8M step, the baselines' performance on the validation set has been decreasing. After trading off training time and computational resources, we chose 1.8M as the training step size for RL. While further training might yield improvements, we argue that the chosen step size provided a reasonable balance on computing resources, time, and performance.
>
> **Q4**:  I would suggest including this information in the main paper directly.
>
> A4: Thank you for your suggestion. We will make the necessary changes to the main paper to include a more detailed baseline description. Additionally, we will provide a comprehensive account of the training and testing processes, with other essential information, in the supplementary material.
>
> **Q5**: the difference between Random-object and Random-demand.
>
> A5: Due to the characters limitation, please see **Common Response 2**.
>
> **Q6**: Performance for GPT-3+Prompt* and MiniGPT-4 in Table 1
>
> A6: Thanks for pointing that out. The results for GPT-3+Prompt* and MiniGPT-4 in Table 1 being the same across all scene and instruction settings can be attributed to several factors. Firstly, both GPT-3 and MiniGPT-4 were not  trained on any scene or instruction. As a result, there is no distinction between seen and unseen scenes/instructions for these models.  Secondly, to ensure the reliability of the results, we conducted extensive testing with thousands of episodes using MiniGPT-4 and GPT-3. Over the course of testing, the results stabilized across all four settings, leading to consistent performance values. Lastly, when presenting the results in the table, we rounded the values, which might lead to data that appears the same in the table but might differ at multiple decimal places.

---

> > ### Comment · Reviewer_7yMw · 2023-08-14
> >
> > I would like to thank the authors for their efforts in trying to address my concerns and the ones of other reviewers. It should particularly be noted that many additional experiments were conducted which is highly appreciated.
> >
> > The authors gave reasonable answers to most of my concerns. I would just like to come back to **Q5**: As mentioned by the authors, several reviewers were confused regarding the difference between *Random-object* and *Random-demand* baselines. After reading the answer from reviewers, it seems that this is not a difference in the baseline itself but rather in the task at hand. I feel like this should be made clearer in the paper, and even in Table 1. Indeed, comparing different methods on the same task and the same method on different tasks is very different, and it looks to me like the authors are doing both simultaneously in Table 1, which makes it harder to draw clear conclusions from experimental results in my opinion.
> >
> > However, I still feel like this paper asks interesting questions, is well-written, and involves an important amount of experiments.

---

> > > ### Author Response · Authors · 2023-08-14
> > >
> > > We are glad that our responses help alleviate your concerns. We also thank you for appreciating our additional experiments.
> > >
> > > **Q7**: I feel like this should be made clearer in the paper, and even in Table 1. Indeed, comparing different methods on the same task and the same method on different tasks is very different, and it looks to me like the authors are doing both simultaneously in Table 1, which makes it harder to draw clear conclusions from experimental results in my opinion.
> > >
> > > A7: Thank you sincerely for your insightful suggestion. We highly appreciate your feedback. we will implement your suggestion by separating the contents of object navigation and DDN in Table 1 into two distinct tables. Additionally, we will provide a more comprehensive and detailed explanation of these two different tasks.
> > >
> > > Many thanks again for your response! We hope to hear back if you have further questions!

---

> > > ### Author Response · Authors · 2023-08-20
> > >
> > > Dear reviewer #7yMw
> > >
> > > Thank you very much for your time and effort spent in reviewing our paper. We appreciate your valuable suggestions for our papers.
> > > **As the discussion period is ending soon, we would like to  kindly request that you take into consideration the possibility of adjusting your score.** Please let us know whether you have further concerns. We are sincerely waiting for your response!
> > >
> > > Best wishes,
> > >
> > > Authors of 2423

---

### Official Review · Reviewer_nwVS · 2023-07-06

**Soundness:** 3 good
**Presentation:** 3 good
**Contribution:** 3 good
**Rating:** 6
**Confidence:** 5

**Summary:**

This paper proposes a Demand-driven Navigation (DDN) problem to leverages the user’s demand as the task instruction and prompts the agent to find an object which matches the specified demand. Then the authors proposed a method by learning demand-conditioned object attribute features from LLMs and align them to visual navigation via CLIP. The experiment shows the efficiency of the proposed method. However, I have some concerns about this paper. My detailed comments are as follows.

**Strengths:**

1. This paper proposes a novel Demand-Driven Navigation task to explore the navigation with only the user’s demand as the task instruction. This task is practical and worth more research, especially with the development of open-vocabulary foundation models.

2. The proposed attribute module is interesting and helpful for extracting attributes of objects.

3. This paper provides a method to tackle the DDN task by extracting common sense from LLMs to learn textual attribute features and uses CLIP to align the textual and visual attribute features. The results obviously outperform the baselines.


**Weaknesses:**

1.	One important baseline is missing. The agent could explore the environment using a heuristic algorithm like FBE. At each time step, the agent detects all objects in observation and ask LLM whether these objects can satisfy the human demand.

2.	What are the differences between common sense knowledge and human preferences mentioned in the paper?

3. It is not clear why the results of random-object are different from random-demand. Are they both execute random actions? More explanations are needed.

4.	In Table1, the ZSON-demand performs better the ZSON-object. Does it indicate that CLIP performs better in understanding high-level abstract demand compared to concrete objects? This result seems to conflict with the claim that “CLIP does not perform well on alignment between instructions and objects” in Line 335.

5.	Some related works that try to solve open-vocabulary navigation[1,2] or scene understanding[3] are missed. It would be better to add and discuss them in the related work part for the sake of completeness.

[1] Weakly-Supervised Multi-Granularity Map Learning for Vision-and-Language Navigation, NeurIPS 2022.

[2] Visual Language Maps for Robot Navigation, ICRA 2023.

[3] LERF: Language Embedded Radiance Fields, ArXiv 2023.


**Questions:**

My main concerns are the lack of an important baseline and the analysis of the experimental results.

**Limitations:**

The authors have adequately addressed the limitations

---

> ### Author Rebuttal · Authors · 2023-08-03
>
> REPLY: Thank you! We have clarified some questions, address your concerns, and hope to hear back from you if you have further questions!
>
> **Q1**: One important baseline is missing. The agent could explore the environment using a heuristic algorithm like FBE. At each time step, the agent detects all objects in observation and ask LLM whether these objects can satisfy the human demand.
>
> A1: Thank you very much for your suggestion. We supplement two experiments with FBE as an exploratory module and MiniGPT-4 and GPT-3 as recognition modules with the following results. Due to Rubuttal character limitations, the results are provided in the attached PDF file in **Common Response**. Since ProcThor is a large scene consisting of multiple rooms, the results show that heuristic search is not efficient. Our model structure mimics VTN, a structure that can learn associations between objects, and learns the attribute features of the objects with Contrastive Learning, which in a way improves the search efficiency by using the semantics of the objects.
>
> **Q2**: What are the differences between common sense knowledge and human preferences mentioned in the paper?
>
> A2: Thank you for your valuable commont. In the context of the paper, common sense knowledge refers to the general knowledge and understanding that humans possess about the world and its functioning. This knowledge includes basic facts and principles that are commonly accepted and expected in everyday life. On the other hand, human preferences refer to the individual preferences, desires, and choices that vary from person to person. These preferences can be influenced by personal experiences, cultural background, and subjective judgments, leading to variations in how individuals perceive and prioritize different options. In our DDN dataset, we did not explicitly express personal preferences, but instead added some modifiers to further specify the target category and reflect personal preferences, such as "I want a place to rest" vs "I want a soft place to rest."
>
> **Q3**: It is not clear why the results of random-object are different from random-demand. Are they both execute random actions? More explanations are needed.
>
> A3: Thank you for pointing that out. Indeed, both random-object and random-demand baselines execute actions randomly from the action space provided. However, the key difference lies in the tasks assigned to them and the criteria used to evaluate success, leading to varied results. For random-object, the baseline is given **a specific category of objects** and is tasked with finding an object of that category within the environment. On the other hand, random-demand is given **a demand instruction** and is asked to find an object that satisfies that particular demand. In some scenarios, a single demand instruction may be satisfied by **multiple categories of objects** present in the environment. Due to this broader range of possible successful outcomes, the success criteria for random-demand are more relaxed compared to random-object, resulting in higher success rates for random-demand.
>
> **Q4**: In Table1, the ZSON-demand performs better the ZSON-object. Does it indicate that CLIP performs better in understanding high-level abstract demand compared to concrete objects? This result seems to conflict with the claim that “CLIP does not perform well on alignment between instructions and objects” in Line 335.
>
> A4: Thank you for raising this concern. It is essential to note that ZSON-demand and ZSON-object represent different task settings. In ZSON-demand, the objective is to find the category of objects that fulfills a given demand instruction, where multiple object categories in the scenes may satisfy the demand. In contrast, ZSON-object entails locating a specific object category. The difference in task settings makes it challenging to draw direct comparisons between ZSON-demand and ZSON-object performance.
>
> Regarding why we claim that CLIP does not perform well on alignment between instructions and objects, we need to first explain the difference in testing and task input between ZSON-object and ZSON-demand. Both ZSON-object and ZSON-demand were trained using Image Navigation (task input is CLIP-vision-feature of the target image), but for testing, ZSON-object was tested with Object Navigation (with the task input being CLIP's text feature of the target object's name), while ZSON-demand was tested with the DDN task (with the task input being CLIP's text feature of the demand instruction). ZSON 's motivation is to rely on CLIP's ability to align between text and vision to accomplish zero-shot object navigation. This alignment ability is reflected in the cosine similarity of the features.The average cosine similarity between **object image features** and **object name features** in ZSON-Object is 0.28; whereas the average cosine similarity between **object image features** and **instruction features** in ZSON-demand is only 0.22. So we argue that the alignment between instructions and objects in DDN task is not as good as in Object Navigation task.
>
> **Q5**: Related work
>
> A5: Many thanks for the papers you have provided. We think these papers are very relevant to our work, so we will add them all to the related work section and discuss them.

---

> > ### Comment · Reviewer_nwVS · 2023-08-16
> > **Thanks for Response**
> >
> > Thanks for your detailed response, which solved all my concerns about experimental settings and results analysis. I am happy to raise my score.

---

> > > ### Author Response · Authors · 2023-08-16
> > >
> > > We are glad that our responses help alleviate your concerns.  Thank you for raising your score! We greatly appreciate your valuable suggestions on our paper.

---

### Official Review · Reviewer_JLnR · 2023-07-07

**Soundness:** 3 good
**Presentation:** 4 excellent
**Contribution:** 3 good
**Rating:** 7
**Confidence:** 4

**Summary:**

This paper proposes the novel task of demand-driven navigation, where a robot must navigate towards a goal object that satisfies the human user's demand (e.g., for the demand "I am thirsty", the robot has to find water/juice/tea, etc.) Citing limitations of navigation methods used for other object-goal navigation variants, the paper also proposes a novel architecture that relies on extracting attribute features conditioned on the demand text features and object features for visible objects. The attribute features extracted are indicative of what physical / semantic properties are fulfilled by a given object to satisfy the demand (e.g., the object "water bottle" has the property to "quench thirst"). These features are learned using a contrastive learning objective on prior-knowledge extracted from a large-language model (GPT-3). Experiments on the ProcThor dataset demonstrate the difficulty of the demand-driven navigation task and the superiority of the proposed method over prior navigation methods.

**Strengths:**

* The idea of demand-driven navigation is interesting and novel. While the goal specification is natural language (similar to prior work), it focuses on a very different concept of "demand" where multiple functionally equivalent objects can satisfy a given demand.
* The problem statement is well motivated and the paper writing clarity is largely good (see weaknesses for some questions).
* The supplementary material provides necessary implementation details and details about the dataset for reproducibility.
* The experimental design is good. Recent baselines for open and closed vocabulary navigation are considered. Multiple experimental trials have been performed to show statistical significance. The proposed method significantly outperforms the baselines.

**Weaknesses:**

# Post-rebuttal update

I thank the authors for addressing my questions and concerns in the rebuttal. It is clear to me that this paper presents a valuable new direction in this space of navigation tasks and the experiments are sufficiently strong to recommend acceptance. The motivation can be clarified further and the rebuttal responses need to be reflected in the final paper. With the understanding that this will be done, I am increasing my rating to accept (7).

---------------------------------------------------------------

## Task motivation good, but practical implications are not clear

I liked the task itself, but a key question that concerns me is *"how often does an object demanded not be present in the scene, and therefore, a functionally equivalent object was needed to satisfy the demand?"*. That is, how often do demands for objects become infeasible because the object was missing in the scene? E.g., if the demand is "Get me a water bottle because I am thirsty", I would expect most scenes would contain water bottles. The task definition avoids addressing this issue by having a generic demand itself as the input.

Of course, it is possible that there are other objects that can be functionally equivalent **in addition** to the object demanded, but that's not the scenario motivated in the introduction. Additionally, as a user, if I request a water bottle, that's exactly what I'd want unless it is nowhere to be found. So the proposed task makes more sense in the absence of the primary object.


## Dataset is underwhelming

L63 - 64 - "mapping between demands and objects is many-to-many" ---This is only partly true. Based on statistics in Figure 4 supp., only 2.3 objects, on average, correspond to a given instruction. Calling this "many" is underwhelming. The other direction is still true though, i.e., there are many instructions satisfied by a given object (Figure 5 supp).

## Task definition fails to consider whether an object instance satisfies the demand
L89 - "Both a bottle of water and a cup of tea" can "quench thirst" --- this is only true if the bottle has water and cup is filled with tea. Does the dataset / task differentiate between bottle/cup instances that contain liquids vs. those that are empty?

## Approach clarifications needed
* L175 - "only take RGB images as sensor inputs" --- is the GPS+compass information also included here or does the model learn to localize on its own?
* L185 - why is the time step limit only 100? That seems very short for navigation in large environments.
* L257 - why is the attribute module a transformer (e.g., why not just an MLP)? Is self-attention across demand-object features needed?
* L281 - why use only imitation learning and not reinforcement learning?

## Experiment clarifications needed
* Table 1 - why is the performance on seen scene, seen instruction so low? I'd expect close to 100% success due to overfitting.
* L306 - how are the object categories derived from demand inputs for the *-object methods?
* L317 - why are there variants of "Random" if the policy only selects actions randomly?
* L336 - 338 - "likely to be due to the fact that ... have a high likelihood not to be present ... meaningless search" --- can we empirically quantify this? How often is it the case that objects predicted by GPT-3 are missing in the scene?
* L371 - 373 - "surpasses all baselines ... CLIP-visual features helps ..." - it seems to me an unfair advantage to use CLIP visual features only for the proposed method and not the baselines. How do baselines perform when equipped with CLIP features?

**Questions:**

I'd appreciate it if the authors can clarify the questions raised in weaknesses. It will help me arrive at my final decision.

**Limitations:**

Yes, the limitations are discussed.

---

> ### Author Rebuttal · Authors · 2023-08-08
>
> REPLY: Thank you! We have clarified some questions, address your concerns, and hope to hear back from you if you have further questions!
>
> **Q1**: how often does an object demanded not be present in the scene, and therefore, a functionally equivalent object was needed to satisfy the demand?
>
> A1: Thank you for pointing this out! We carefully analyzed the scenario you pointed out, wherein the Procthor contains multiple objects (e.g., objects A, B, and C) that fulfill the demand instruction, but not all of them are simultaneously present within a specific scene. Our statistical calculations revealed a probability of 54.4% for this situation to occur. Correspondingly, there is a 45.6% chance that all objects satisfying the demand instruction are indeed present in the scene.
>
> **Q2**: Additionally, as a user, if I request a water bottle, that's exactly what I'd want unless it is nowhere to be found.
>
> A2: Thank you very much for your valuable suggestion. Due to Rebuttal character limitations, we put the relevant discussions into **Common Response 3**. we would appreciate it if you could read them.
>
> **Q3**: Dataset is underwhelming. only 2.3 objects, on average, correspond to a given instruction
>
> A3: Thank you very much for pointing this out. We argue that the nature of DDN tasks is many-to-many. We acknowledge that the limited number of object categories available in the ProcThor dataset might have influenced the presentation of this nature, resulting in an average of only 2.3 object categories per instruction being fulfilled. In real life, a demand can be satisfied by a much larger number of object categories, for example, we can easily generate 10 different categories of objects for each instruction using GPT-3 ( please see language-grounding mappings in supp 8.2.1). In the future work, we will focus on more diverse scenes and objects to generate a more complex DDN dataset.
>
> **Q4**: Task definition fails to consider whether an object instance satisfies the demand. Does the dataset / task differentiate between bottle/cup instances that contain liquids vs. those that are empty?
>
> A4: Thank you for raising this important point. Yes, the dataset differentiates them.  For instance, in the demand "I am thirsty," the required object is "water," rather than specifying a particular container like "bottle" or "cup." On the other hand, in the demand "I need a container to hold water," the required object is explicitly defined as "bottle." We can identify them using the simulator's metadata easily.
>
> **Q5**: Approach clarifications
>
> L175:
>
> No, because many previous visual navigation works have used depth and GPS, we want to emphasize that we only used RGB.
>
> L185:
>
> Our decision to set the time step limit to 100 is based on the analysis of our traject dataset. The dataset, collected using the A* algorithm, revealed that 90% of the trajectory lengths are less than 50 and the average length is 27.24. Consequently, we opted for a time step limit that is twice times longer than 50.
>
> L257:
>
> Thank you for your insightful question and valuable suggestions. We chose to implement the attribute module as a transformer because transformers have demonstrated remarkable effectiveness in various domains, such as NLP and computer vision.
> To show the transformer's ability for learning attribute features, we added experiments by replacing the transformer with a 4-layer MLP in the attribute module.  **Due to Rubuttal character limitations, the results of the experiment are shown in the attached PDF in Common Response.** The results clearly showed that the MLP was not superior to the transformer in learning attribute features. Thus, we found that self-attention across demand-object features, provided by the transformer, is essential for achieving optimal performance in our task.
>
> L281:
>
> Thank you for your insightful suggestion! We agree that integrating our method with reinforcement learning (RL) could potentially lead to further performance improvements, conceptually. However, due to the size of our model (even several times larger than VTN), the reward signal from RL turns out to be weak, rendering RL less effective for our method. Consequently, we initially focused on exploring an imitation learning (IL)-based method, which yielded significant performance gains over baselines. Nonetheless, we recognize the value and promise of integrating RL into our method as a valuable future direction for further research.
>
> **Q6**:Experiment clarifications
>
> Table 1:
>
> The performance that is not 100% on seen scenes and instructions can be attributed to the limited size of our trajectory dataset. We only collect up to 3 trajectories with different initial positions for each instruction and each room. Since each room contains hundreds or even thousands of initial positions, each corresponding to different final objects found, the trajectory dataset we collect is relatively small compared to the vast number of all possible trajectories, even less than 1%. This limited dataset size makes it challenging for our model to overfit to the specific seen scenes and instructions, resulting in performance below the expected 100% success rate.
>
> L306:
>
> The object categories are obtained by asking GPT-3 for the current demand instruction and letting GPT-3 provide the categories of objects that fulfill the demand. Subsequently, we convert the GPT-3 generated answers into formats acceptable by VTN and ZSON, respectively, using different methods. For detailed information on the conversion process, please refer to the supplementary materials Section 8.3.2.
>
> L317:
>
> Please see **Common Response 2**.
>
> L336-338:
>
> As we described in Q1, there is a 54.4% probability that the object given by GPT-3 does not exist in the current scene, and there are other objects that satisfy the given demand instruction.
>
> L371-373:
>
> Thank you for your suggestion. We added the experiments VTN+CLIP-demand. The results are in the attached PDF file in **Common Response**.

---

> > ### Author Response · Authors · 2023-08-15
> > **Some clarifications on Q6 L306**
> >
> > We apologize that we misunderstood your question on the Q6 L306 at first.  Here are some clarifications.
> >
> > Methods with the -object suffix are trained in **visual object navigation** and tested in **visual object navigation**. Instead of deriving from demand inputs, we directly inform the robot of the object category that the robot needs to find. We apologize for putting them together with the demand-driven navigation results that caused some misunderstanding. We will make a separate table for methods with -object suffix later.
> >
> > Methods with the -GPT suffix were used with models trained in **visual object navigation** and tested in **demand-driven navigation**. We get an target object category that can satisfy the demand instruction by asking GPT-3. Then we inform the robot of the target object category given by GPT-3.
> >
> > We will also explain the different suffixes in more detail in the main paper.

---

> > > ### Comment · Reviewer_JLnR · 2023-08-17
> > > **Reviewer response to rebuttal**
> > >
> > > I thank the authors for their detailed rebuttal. It clarified a majority of my concerns, so I feel more positively about the paper. I have one outstanding concern that was not addressed completely, so I will re-state my concern here. This is related to Q1 and Q2. I will split my concern into two parts for clarity.
> > >
> > > ## Part 1: Are there statistical biases in the data?
> > >
> > > It is not a question of whether all objects satisfying a demand are jointly present in the scene. Rather, does one particular object satisfying the demand always be present in every scene (e.g., to satisfy the demand “I want to quench my thirst”, is a “water bottle” or alternatively a “cup” always present in any given scene?). Does the agent have to reason about alternative objects at all to satisfy a given demand? For example, it might just have to deterministically learn a one-to-one mapping from demand -> object. This would be a form of a statistical bias in the data.
> > >
> > > ## Part 2: Can the authors clarify the motivation of “demand-driven navigation” further?
> > >
> > >  It is common for humans to ask for certain objects that satisfy their demand, as opposed to just making a demand (e.g., get me a water bottle because I’m thirsty). If that particular object is present in the scene, then that is exactly what a human user would expect to get (not an alternate object that satisfies the demand). For instance, if I asked a robot to get me my water bottle since I was thirsty, I would be fairly annoyed if I got water in a cup. I might have asked for a water bottle because I additionally wanted to take it with me to the office. I shouldn’t have to convey my intentions completely to a robot just to get a water bottle.
> > >
> > > I’m looking for examples that can very clearly motivate the proposed task. My impression is that for some common demands, it is very unlikely that an agent has to think about more than one object category that satisfies the demand (even if those objects may be present in the scene). “I am thirsty” —> always get water bottle, “I want to sit down” —> always find a chair, etc. Having good examples here is critical to attract attention to this task.

---

> > > > ### Author Response · Authors · 2023-08-17
> > > >
> > > > We're glad our responses help alleviate your concerns. Thank you very much for your response. Regarding your further questions, our responses are as follows.
> > > >
> > > > **Q7**: does one particular object satisfying the demand always be present in every scene (e.g., to satisfy the demand “I want to quench my thirst”, is a “water bottle” or alternatively a “cup” always present in any given scene?). Does the agent have to reason about alternative objects at all to satisfy a given demand? For example, it might just have to deterministically learn a one-to-one mapping from demand -> object.
> > > >
> > > > A7: If a demand instruction can be satisfied by only one object, then that object must exist in the scene during training. We took into account demand instructions that can be satisfied by the existence of multiple objects. However, only **6.5%** of the multi-object demand instructions align with the situations you described, where a particular object always satisfies the given demand instruction in the training scene. This indicates that, in the vast majority of cases regarding multi-object demand instructions, the agent must reason based on the context of the current scene rather than rigidly searching for a fixed object. On average, we statistically find that objects satisfying a given multi-object demand instruction have a **50.48%** probability of appearing within the current scene during training.
> > > >
> > > > **Q8**: It is common for humans to ask for certain objects that satisfy their demand, as opposed to just making a demand (e.g., get me a water bottle because I’m thirsty). If that particular object is present in the scene, then that is exactly what a human user would expect to get (not an alternate object that satisfies the demand). For instance, if I asked a robot to get me my water bottle since I was thirsty, I would be fairly annoyed if I got water in a cup. I might have asked for a water bottle because I additionally wanted to take it with me to the office.
> > > >
> > > > A8: One important reason why we use demand instructions as task descriptions is that the natural language-based demand instruction description space is very wide. We can express our demands well in natural language and then let agent satisfy our demands. In the case of your example, it could be presented in a more refined manner, such as "I'm a little thirsty and will be going to my office. Can you help me find a thirst-quenching object that is also portable for me to carry?"
> > > >
> > > > We understand and agree that if the desired object is present in the scene, it should be prioritized over alternative objects that may also satisfy the demand. In future work, we will consider incorporating prioritization of objects based on user preferences. For example, a demand instruction like "I'm thirsty, please make it a priority to find me a bottle of water" would explicitly prioritize finding a bottle of water.
> > > >
> > > >
> > > > **Q9**: I’m looking for examples that can very clearly motivate the proposed task. My impression is that for some common demands, it is very unlikely that an agent has to think about more than one object category that satisfies the demand (even if those objects may be present in the scene). “I am thirsty” —> always get water bottle, “I want to sit down” —> always find a chair, etc. Having good examples here is critical to attract attention to this task.
> > > >
> > > > A9: Thank you for your feedback and suggestion. We understand the importance of providing clear examples that can effectively motivate the proposed task. Consider the demand "I'm cold." It may seem intuitive that a single object category like "clothes" can satisfy this demand. However, along with "clothes," the agent could also consider objects like "a blanket," "a cup of hot coffee," or even "an air conditioner" in certain scenarios.
> > > >
> > > >
> > > > Many thanks again for your response! We hope to hear back if you have further questions!

---

> > > > ### Author Response · Authors · 2023-08-20
> > > >
> > > > Dear reviewer #JLnR
> > > >
> > > > Thank you very much for your time and effort spent in reviewing our paper. We appreciate your valuable suggestions for our papers.
> > > > **As the discussion period is ending soon, we would like to  kindly request that you take into consideration the possibility of adjusting your score.** Please let us know whether you have further concerns. We are sincerely waiting for your response!
> > > >
> > > > Best wishes,
> > > >
> > > > Authors of 2423

---

> > > > > ### Comment · Reviewer_JLnR · 2023-08-21
> > > > > **Reviewer response to rebuttal (part 2)**
> > > > >
> > > > > I thank the authors for addressing the potential biases in the data and clarifying the motivation. I feel it is worth addressing the motivation of the task, i.e., how it pertains to a specific aspect of the task, and more things can be done in the future, i.e., considering the human preference for particular objects, etc. in the paper (possibly in the limitations section). This doesn't take anything away from the merits of the paper, but it does show the community that these next steps are important to take in the future.
> > > > >
> > > > > I'm happy to increase my rating to Accept (7) and willing to push for final acceptance.

---

> > > > > > ### Author Response · Authors · 2023-08-21
> > > > > >
> > > > > > We are glad that our responses have helped alleviate your concerns. Thank you for raising your score! Your valuable suggestions on our paper and your willingness to push for the final decision are greatly appreciated.

---

### Author Rebuttal · Authors · 2023-08-03

## Common Response ##
We thank all reviewers for appreciating our DDN task, method and experiments.  "The idea of demand-driven navigation is interesting and novel." (JLnR) "The proposed attribute module is interesting and helpful for extracting attributes of objects." (nwVS) "Authors propose diverse baselines to evaluate the performance of current language models in robotics/navigation tasks." (7yMw) "The motivation of this work is interesting." (i2zg) "The paper flows nicely with good motivation of the novel task and its challenges clearly stated. The proposed method makes sense and comparison is good." (FyRT)

**(Common Response 1)** However, we notice that some reviewers (JLnR, nwVS, 7yMw, i2zg) have some suggestions about our experiments, such as missing some important baselines and ablation experiments. We supplement with five baselines (FBE+MiniGPT-4, FBE+GPT-3, CLIP-Nav+MiniGPT-4, CLIP-Nav+GPT-3, VTN+CLIP) and two ablation experiments (Ours_w/o_BERT, Ours_w/o_Attribute_Transformer) as they suggested. The original ablation Ours_w/o_attr in the main paper is renamed Ours_w/o_Attribute_Pretrain. Due to time and computational resource constraints, we conducted only one round of each experiment. We will conduct more rounds of experiments using different random seeds if the paper is accepted. **The supplementary experimental results are shown in the attached PDF file in Common Response**.

**(Common Response 2)** We also notice that some reviewers (JLnR, nwVS, 7yMw) have some questions on our Random baselines. The inclusion of variants of "Random" in our baselines is motivated by previous research [1,2,3], where "Random" is used as a baseline to reflect the task's difficulty.
Random-object and Random-demand are different in terms of task settings. Random-object's task is to find **a given object category** whereas Random-demand's task is to find an object that satisfies a given demand instruction. In some scenarios, a single demand instruction may be satisfied by **multiple object categories** present in the environment. Due to this broader range of possible successful outcomes, the success criteria for Random-demand are more relaxed compared to Random-object, resulting in higher success rates for random-demand. Therefore, theoretically the result of Random-demand should be better. However, because these two baselines have different task settings, it is difficult to draw valuable conclusions from a comparison between them.

**(Common Response 3)** Some reviewers (JLnR, i2zg, FyRT) have valuable suggestions on the content of our DDN dataset. They suggested that we could add more fine-grained demand instructions such as "I am thirsty but I cannot drink cold", "something quench thirst and contains caffeine". We have actually included some similar examples in our DDN dataset, such as "I need a place to rest" and "I need a soft place to rest". The former can be "bed, wooden chair, sofa" while the latter cannot be "wooden chair" but "bed, sofa". The latter contains two demands in its instruction: "soft" and "available for rest". In future work, we'll take their suggestions into serious consideration and add some explicit personal preferences as well as more complex demand instructions, including negation of object samples, combining multiple demands, and prioritising between objects.

We truly appreciate the time all the reviewers, AC and SAC have taken to carefully review our work. What follows are the revision plan and  point-to-point responses, and we hope that our responses addresses your concerns. In the attached PDF file, we summarise the results of all experiments (both baseline and ablation experiments) including the experiments we supplemented in the rebuttal period. Thanks again for all valuable comments and suggestions.


## Revision Plan ##
As suggested by reviewer i2zg, we will reduce some of the redundant expressions in the Introduction section to make it more concise and focused.

In the Related Work section, we will add some papers recommended by reviewer nwVS and discuss them.

In the Experiment section, we will add the results of our supplementary baseline and ablation experiments to Table.1. We wiil also add more detailed description about baselines.

### References ###

[1] Du, H., Yu, X., & Zheng, L. (2021). VTNet: Visual transformer network for object goal navigation. ICLR 2021

[2] Anderson, P., Wu, Q., Teney, D., Bruce, J., Johnson, M., Sünderhauf, N., ... & Van Den Hengel, A. (2018). Vision-and-language navigation: Interpreting visually-grounded navigation instructions in real environments. CVPR 2018

[3] Chen, C., Jain, U., Schissler, C., Gari, S. V. A., Al-Halah, Z., Ithapu, V. K., ... & Grauman, K. (2020). Soundspaces: Audio-visual navigation in 3d environments.  ECCV 2020

---

### Comment · Area_Chair_zeLu · 2023-08-11
**Discussion with authors**

Dear Reviewers,

Please check the rebuttal and start a discussion with authors if you need any additional information to make your final decision. The discussions should be completed by Aug 16.

Thanks,

AC

---

### Decision · Program_Chairs · 2023-09-21

**Decision:**

Accept (poster)

**Comment:**

The reviewers initially had some concerns regarding practical implications, missing baselines and ablations, clarity of experiments and concepts, connection with object-goal navigation, etc. These aspects were comprehensively addressed in the rebuttal, ultimately resulting in an increase in the ratings.

Reviewer i2zg recommends rejection while the others are on the accept side. While the concerns from Reviewer i2zg are valid and the insights are valuable, the AC believes the provided rebuttal alleviates the concerns. Therefore, the AC follows the recommendation of the majority of the reviewers and recommends acceptance.